# Perception and practice of self-medication with antibiotics among medical students in Sudanese universities: A cross-sectional study

**Osman Kamal Osman Elmahi**[1]*, **Reem Abdalla Elsiddig Musa**[2], **Ahd Alaaeldin Hussain Shareef**[3], **Mohammed Eltahier Abdalla Omer**[3], **Mugahid Awad Mohamed Elmahi**[4], **Randa Ahmed Abdalrheem Altamih**[5], **Rayan Ibrahim Hamid Mohamed**[5], **Tagwa Faisal Mohamed Alsadig**[6]

1 Faculty of Medicine, Ibn Sina University, Khartoum, Sudan, 2 Faculty of Medicine and Health Sciences, University of Bahkt Alruda, Ed Dueim, Sudan, 3 Faculty of Medicine and Health Sciences, University of Gadarif, Gadarif, Sudan, 4 Faculty of Medicine, University of Gezira, Wad Madani, Sudan, 5 Faculty of Medicine, University of Khartoum, Khartoum, Sudan, 6 Department of Community Medicine, Faculty of Medicine, Alzaiem Alazhari University, Khartoum, Sudan

* osman19091995@gmail.com

## Abstract

### Introduction

The benefits of antibiotics are under threat by self-medication, which culminated in economic burdening of developing countries, treatment failures, the emergence of antibiotic-resistant strains of bacteria and an increased probability of exposure and infection of the general population by antibiotic-resistant bacterial strains.

### Objectives

This study aimed to evaluate the knowledge and attitude of medical students in Sudan towards the use of antibiotics, the prevalence of self-medication with antibiotics among medical students in Sudan and to identify risk factors which promote self-medication with antibiotics.

### Materials and methods

This was a cross-sectional, descriptive and institution-based study, between November 2020 and May 2021. 1,110 medical students were selected by multistage cluster sampling. Logistic regression was used to identify risk factors of self-medication with antibiotics among the study participants.

### Results

The median knowledge score was 7 out of a maximum of 10 (IQR: 5–8). A moderately positive attitude was observed among the participants (Median: 7/10; IQR: 6–8). Knowledge and attitude scores were significantly associated with academic year and monthly allowance (p < 0.05). 675 (60.8%) self-medicated with antibiotics within the previous 12 months, mostly from community pharmacies (321/675; 47.5%). Antibiotics were most commonly used to

**Data Availability Statement:** The datasets used and/or analysed during the current study are

available from the following DOI link: https://doi.org/10.6084/m9.figshare.17695865.

**Funding:** The authors received no specific funding for this work.

**Competing interests:** The authors have declared that no competing interests exist.

treat respiratory tract infections (38.1%) and cough (30.4%). Chi-square analysis demonstrated that self-medication with antibiotics was significantly associated with gender, year of study and monthly income.

## Conclusions

Undergraduate medical students had moderate knowledge and attitude towards antibiotic use and antibiotic resistance, and an alarmingly high prevalence of self-medication with antibiotics. This highlights the urgent need for tighter legislation regarding the sales of antibiotics in community pharmacies by the state and federal health ministries.

## Introduction

Antibiotics are medications which inhibit the growth of bacteria and are prepared for prevention and treatment of bacterial diseases [1]. Since the introduction of antimicrobial agents as part of an essential drugs concept developed by the World Health Organization (WHO) in the 1970s, antibiotic stewardship has played a significant role in improving public health [2, 3]. These medications are essential treatments in the developing world, particularly where infectious diseases remain a common cause of morbidity and mortality [4]. However, in the developing world, the societal, economic and therapeutic benefits of antibiotics have come under threat by self-medication and parent-to-child medication, usually with over-the-counter antibiotics [5–9]. Self-medication can be described as the use of medications to treat self-diagnosed disorders or symptoms, or the intermittent use of a prescribed drug for chronic or recurrent diseases or symptoms [10]. Various practices of self-medication include the acquisition of these medications without prescription from a certified medical practitioner, reuse of previous prescriptions, use of leftover medications and sharing with family members, neighbours and others [5, 11]. Reasons for practicing self-medication are unique to each region and can be related to various factors including health systems, location [12], poverty, gender and age [13]. In some regions, self-medication is regarded as a gateway to independence from health systems [14] and a means of observing a human right to refuse treatment from physicians [15]. Self-medication with antibiotics is directly linked to inappropriate use, such as inadequate dosage, sharing of medications and cessation of treatment with cessation of symptoms [16]. This leads to an increased risk of drug interactions, the concealing of symptoms of underlying conditions and the emergence of antibiotic-resistant strains [17]. Furthermore, the outcome of improper use of antibiotics has been documented as an increased incidence of water-borne and food-borne infections by antibiotic-resistant bacteria, nosocomial infections and suppressed animal production. As a result of the advent of resistant strains, prolonged hospital stays due to failure of treatment regimens, in conjunction with increased community movement, culminates in a greater risk of the general population to contracting strains of antibiotic-resistant bacteria [18, 19]. The overall mortality rate of antimicrobial resistance is currently estimated at 700,000 lives per annum and is projected to rise to 10 million lives annually by 2050. In an effort to combat the resultant increase in mortality, the World Health Organization has initiated a global effort to raise awareness of antimicrobial resistance and increase antibiotic stewardship [20].

In Sudan, studies have shown alarmingly high prevalence rates of self-medication with antibiotics in Khartoum State, with prevalence rates ranging between 41% [21] to 79.5% [22] and the practice being significantly associated with age [21, 22] and income [10]. Antibiotics in

Sudan are readily accessible without the need for prescriptions, with little control of sales by governing bodies, which subsequently leads to an increased economic burdening on state and federal health systems [23–25].

The aim of this study was to investigate the knowledge and attitude of undergraduate medical students in Sudan, in eight universities across five states, towards antibiotic use and antibiotic resistance. Also, this study sought to assess the prevalence of self-medication with antibiotics among the participants, the risk factors which promote practicing self-medication with antibiotics, the most common symptoms treated by antibiotic self-medication and the reasons for practicing self-medication with antibiotics. This study was motivated by the recent change in population movement between various states in Sudan, the absence of data on self-medication with antibiotics at a national level in Sudan and the alarming spread of antibiotic-resistant strains of bacteria through through Sudan, Africa and the Eastern Mediterranean Region.

## Methodology

### Study design

This was a descriptive, cross-sectional study conducted in the medical schools of University of Khartoum, Alzaiem Alazhari University, University of Gezira, University of Gadarif, University of Bakht Alruda, National Ribat University, Omdurman Islamic University and Ahfad University for Women between November 2020 and May 2021. The total population of medical students was 12,350 people.

### Sampling size and technique

A pilot study was conducted, in which the prevalence of non-prescription antibiotic use was 49%. The calculated sample size was 1,110 medical students according to the mathematical equation below, considering p = 0.49, Z = 1.96 at 95% confidence and N = 12,350. The response rate was 100%.

$$\frac{\frac{z^2 \times p(1-p)}{e^2}}{1 + \left(\frac{z^2 \times p(1-p)}{e^2 N}\right)}$$

A multistage cluster sampling technique was applied to proportionally select the participants. In the first stage, medical students from each university were selected. The second stage involved the acquisition of a simple random sample of medical students from each academic year, based on ID numbers registered at the Deanship of Student Affairs in each medical school.

### Data collection tools and methods

A closed-ended, pre-validated and pretested questionnaire was used to collect data from the respondents, consisting of 38 questions divided into five sections: Sociodemographic characteristics, Knowledge, Attitude, Practice and Storage.

The first section recorded the socio-demographic characteristics of the respondents including gender, academic year, monthly allowance, health insurance status and relatives working in the health sector.

The knowledge and attitude sections of the questionnaire each contained 10 questions whose responses were measured with a five-point Likert scale. One point was awarded for the correct answer (answering "Agree/Strongly Agree" to correct statements and items of positive

attitude and "Disagree/Strongly Disagree" to incorrect statements and items of negative attitude), while no points were awarded for "Not Sure" and the incorrect responses. For quantitative analysis of the responses to knowledge, a score of <60% was considered poor, 60–80% was considered moderate and ≥80% was considered a good score in both sections.

In the practice section of the questionnaire, participants were asked if non-prescription antibiotics were taken within the previous 12 months and obliged to answer five further questions regarding their source, symptoms treated, the reasons for self-medicating with antibiotics and the name of the antibiotics taken. All respondents were required to answer four further questions on how to use antibiotics. The storage section contained three questions on whether antibiotics are stored at home, how often they are stored and where respondents store antibiotics. Responses to these sections were not given a score.

## Ethics

Ethical clearance was obtained from the Medical Research Ethics Committee of the Faculty of Medicine and Health Sciences, University of Gadarif. The objectives of the study, beneficence and non-maleficence were explained to the respondents prior to obtaining informed consent to participate in our study.

## Data analysis

The data collected was cleaned, analysed and presented using SPSS version 26.0.0 for macOS Catalina. Responses on sociodemographic characteristics were presented using frequency and percentages. As the data on knowledge and attitude scores were not normally distributed, Wilcoxon rank testing was applied to evaluate associations between median scores and the respondents' characteristics. Chi-square testing was also used to identify risk factors of non-prescription antibiotic use among the participants. A P-value of < 0.05 was considered significant.

## Results

### Sociodemographic characteristics

The majority of the study population were females (69.2%). 29.7% were fourth-year medical students and 32.4% were receiving monthly allowances of 5,000–10,000 SDG (75–150 USD) per month. Students with health insurance accounted for 49.6% (Table 1).

### Knowledge

A median knowledge score of 7/10 (IQR: 5–8) was achieved by the respondents in this study, thus demonstrating a moderate level of knowledge towards antibiotic use. 329 (29.6%) achieved good knowledge scores, while 484 (43.6%) achieved moderate knowledge scores. 297 (26.7%) of the respondents showed poor knowledge. Median scores were significantly associated with academic year and monthly allowance (P < 0.01; Wilcoxon rank test). 6th-year medical students had significantly higher knowledge on antibiotics than medical students in other years (Table 1).

The majority of students in this study (943/1110; 84.9%) were aware that repetitive non-compliance with antibiotics leads to antibiotic resistance, while one-fifth (204/1110; 18.3%) incorrectly stated that cold and flu symptoms can be treated with antibiotics (Table 2).

**Table 1. Sociodemographic characteristics of the respondents and median knowledge and attitude scores.**

| Sociodemographic Characteristics | n (%) | Knowledge score: Median (IQR) | P (Wilcoxon sum test) | Attitude score: Median (IQR) | P (Wilcoxon sum test) |
|---|---|---|---|---|---|
| **Gender** | | | **0.1** | | **0.168** |
| **Male** | 342 (30.8%) | 7(5–8) | | 6(4–8) | |
| **Female** | 768 (69.2%) | 7(5–8) | | 6(5–8) | |
| **Academic Year** | | | **< 0.001** | | **0.003** |
| **First Year** | 221 (19.9%) | 6(5–7) | | 6(4–8) | |
| **Second Year** | 159 (14.3%) | 6(5–6) | | 6(4–7) | |
| **Third Year** | 206 (18.6%) | 7(5–8) | | 6(4–7) | |
| **Fourth Year** | 330 (29.7%) | 7(6–8) | | 6(5–8) | |
| **Fifth Year** | 143 (12.9%) | 7(6–8) | | 6(5–8) | |
| **Sixth Year** | 51 (4.6%) | 8(7–9) | | 7(6–8) | |
| **Monthly Allowance** | | | **0.046** | | **0.002** |
| **<5,000 SDG** | 290 (26.1%) | 6.5(5–8) | | 6(5–7) | |
| **5,000–10,000 SDG** | 360 (32.4%) | 7(6–8) | | 6(5–8) | |
| **10,000–15,000 SDG** | 209 (18.9%) | 6(6–8) | | 7(5–8) | |
| **>15,000 SDG** | 251 (22.6%) | 7(5–8) | | 6(5–8) | |
| **Family members in the health field** | | | **< 0.001** | | **0.009** |
| **Yes** | 691 (62.3%) | 6(5–8) | | 6(5–8) | |
| **No** | 419 (37.7%) | 7(5–8) | | 6(5–7) | |
| **Health insurance** | | | **0.137** | | **0.096** |
| **Yes** | 551 (49.6%) | 6(5–8) | | 6(5–8) | |
| **No** | 559 (50.4%) | 7(6–8) | | 6(5–8) | |

## Attitude

The median attitude score achieved by medical students in this study was 7/10 (IQR: 6–8). Wilcoxon rank testing depicted that attitude scores varied significantly among the participants according to academic year, monthly allowance and having relatives in the health field (P < 0.01).

(Table 1). Over three-fifths of medical students (63.6%) would ask their pharmacist or physician for antibiotics, while 14.1% held the misconception that more expensive antibiotic regimens had less side effects (Table 3). Overall attitude scores demonstrated a moderately positive correlation to knowledge scores in this study (r = 0.441; Spearman's rho: 0.403) and a statistically significant relationship (p < 0.001).

**Table 2. Responses to questions related to knowledge (n = 1,110).**

| | Strongly Disagree: n (%) | Disagree: n (%) | Not Sure: n (%) | Agree: n (%) | Strongly Agree: n (%) |
|---|---|---|---|---|---|
| Antibiotics can be used to treat cold and flu symptoms. | 561 (50.5%) | 148 (13.3%) | 197 (17.7%) | 76 (6.8%) | 128 (11.5%) |
| Antibiotics should not be purchased without prescriptions. | 122 (11.0%) | 68 (6.1% | 139 (12.5%) | 156 (14.1%) | 625 (56.3%) |
| It is safe to use any antibiotic during pregnancy. | 619 (5.8%) | 214 (19.3%) | 215 (19.4%) | 47 (4.2%) | 15 (1.4%) |
| Antibiotics will always be effective in treating the same diseases in future. | 640 (57.7%) | 257 (23.2%) | 88 (7.9%) | 61 (5.5%) | 64 (5.8%) |
| Allergic reactions can occur from using antibiotics. | 22 (2.0%) | 65 (5.9%) | 164 (14.8%) | 253 (22.8%) | 606 (54.6%) |
| Different infections require separate antibiotic treatment regimens. | 214 (19.3%) | 81 (7.3%) | 214 (19.3%) | 271 (24.4%) | 330 (29.7%) |
| Antibiotics can be used to treat bilharziasis. | 394 (35.5%) | 160 (14.4%) | 319 (28.7%) | 119 (10.7%) | 118 (10.6%) |
| Most sore throats are treated with antibiotics. | 275 (24.8%) | 182 (16.4%) | 303 (27.3%) | 222 (20.0%) | 128 (11.5%) |
| It is preferable to use broad-spectrum antibiotics under all circumstances. | 404 (36.4%) | 221 (19.9%) | 260 (23.4%) | 144 (13.0%) | 81 (7.3%) |
| Repetitive non-compliance with antibiotic courses may lead to antibiotic resistance. | 19 (1.7%) | 31 (2.8%) | 117 (10.5%) | 167 (15.0%) | 776 (70.0%) |

## Practice

Of the 1,110 medical students, 675 (60.8%) had self-medicated with antibiotics within the previous 12 months (Table 4). Chi-square testing demonstrated that the use of non-prescription antibiotics was significantly associated with gender (P < 0.01), monthly allowance (P = 0.011) and academic year (P < 0.02). Logistic regression analysis also depicted that females were 1.52 times more likely to self-medicate with antibiotics than males (95% CI: 1.16–1.99). Medical students without relatives working in the health field were less likely to use non-prescription antibiotics (OR: 0.83; 95% CI: 0.65–1.07).

The most common source of non-prescribed antibiotics in this study was community pharmacies (47.5%), followed by personally-known physicians (46.1%) (Table 5).

The most common antibiotic used without prescription in this study was found to be azithromycin (202/675; 29.9%), followed by amoxicillin/clavulanic acid (181/675; 26.8%) and erythromycin (87/675; 12.9%). Over one-quarter of the respondents did not recall the

**Table 3. Responses to questions related to attitude (n = 1,110).**

| | Strongly Disagree n (%) | Disagree n (%) | Not Sure n (%) | Agree n (%) | Strongly Agree n (%) |
|---|---|---|---|---|---|
| I believe it is better to self-medicate with antibiotics for minor illnesses than to see a doctor. | 453 (40.8%) | 200 (18.0%) | 173 (15.6%) | 148 (13.3%) | 136 (12.3%) |
| I would only purchase antibiotics with a valid prescription. | 140 (12.6%) | 117 (10.5%) | 232 (20.9%) | 201 (18.1%) | 420 (37.8%) |
| I believe that more expensive antibiotics have fewer side effects. | 505 (45.5%) | 196 (17.7%) | 253 (22.8%) | 102 (9.2%) | 54 (4.8%) |
| I would stop my treatment with antibiotics when my symptoms stop. | 727 (65.5%) | 144 (13.0%) | 102 (9.2%) | 49 (4.4%) | 88 (7.9%) |
| I would use antibiotics for the same reasons as paracetamol. | 579 (52.2%) | 226 (20.3%) | 191 (17.2%) | 64 (5.8%) | 50 (4.5%) |
| I prefer to be prescribed intravenous antibiotics. | 462 (41.6%) | 243 (21.9%) | 302 (27.2%) | 66 (5.9%) | 37 (3.3%) |
| I would use antibiotics to treat malaria. | 260 (23.4%) | 183 (16.5%) | 314 (28.3%) | 179 (16.1%) | 174 (15.7%) |
| I believe it is important to read the leaflet provided. | 19 (1.7%) | 34 (3.1%) | 132 (11.9%) | 213 (19.2%) | 712 (64.1%) |
| I agree that antibiotic resistance can occur from inappropriate use. | 68 (6.1%) | 36 (3.2%) | 172 (15.5%) | 145 (13.1%) | 689 (62.1%) |
| I would ask my pharmacist or doctor for antibiotics. | 116 (10.5%) | 76 (6.8%) | 212 (19.1%) | 203 (18.3%) | 503 (45.3%) |

**Table 4. Sociodemographic characteristics of the respondents and self-medication with antibiotics.**

| Sociodemographic Characteristics | Total (n = 1,110) | Self-medicated with antibiotics (n = 675) | OR | 95% CI | P |
|---|---|---|---|---|---|
| **Gender** | | | | | **0.002** |
| Male | 342 | 231 | 1 | - | |
| Female | 768 | 444 | 1.52 | 1.16–1.99 | |
| **Academic Year** | | | | | **0.002** |
| First Year | 221 | 127 | 1 | - | |
| Second Year | 159 | 81 | 1.3 | 0.86–1.96 | |
| Third Year | 206 | 127 | 0.84 | 0.57–1.24 | |
| Fourth Year | 330 | 226 | 0.62 | 0.44–0.89 | |
| Fifth Year | 143 | 89 | 0.82 | 0.53–1.26 | |
| Sixth Year | 51 | 25 | 1.41 | 0.76–2.59 | |
| **Monthly Allowance** | | | | | **0.011** |
| <5,000 SDG | 290 | 183 | 1 | - | |
| 5,000–10,000 SDG | 360 | 194 | 1.46 | 1.07–2.01 | |
| 10,000–15,000 SDG | 209 | 142 | 0.81 | 0.55–1.18 | |
| >15,000 SDG | 251 | 156 | 1.04 | 0.73–1.48 | |
| **Family members in the health field** | | | | | 0.155 |
| Yes | 691 | 409 | 1 | - | |
| No | 419 | 266 | 0.83 | 0.65–1.07 | |
| **Health insurance** | | | | | 0.533 |
| Yes | 551 | 330 | 1 | - | |
| No | 559 | 345 | 0.93 | 0.73–1.18 | |

antibiotics taken prior to the study (Table 6). The most common symptoms treated with non-prescription antibiotics were upper respiratory tract symptoms (257/675; 38.1%), followed by cough (205/675; 30.4%) and common cold (177/675; 26.2%).

It was found that over three-fifths (682/1110; 61.4%) carefully read the leaflet provided, while 44.3% (492/1110) always complete the course of antibiotics (Table 7).

The major reason cited by the respondents for self-medicating with antibiotics was previous experiences with similar symptoms (335/675; 49.6%). Costly appointments were also common reason to practice antibiotic self-medication in this study (138/675; 20.4%) (Table 7).

When using antibiotics, over three-fifths (682/1110; 61.1%) carefully read the leaflet before consumption. 25.0% (278/1110) of the respondents always store antibiotics at home for future use.

(Table 8) Over two-thirds (759/1110; 68.4%) stored antibiotics during the study. Logistic regression analysis demonstrated that self-medication with antibiotics was significantly associated with storage (P < 0.001) (Table 9).

**Table 5. Sources of non-prescription antibiotics (n = 675).**

| Source of non-prescription antibiotics (n = 675) | n (%) |
|---|---|
| Community pharmacies | 321 (47.5%) |
| Personally-known physicians | 311 (46.1%) |
| Friends from abroad | 24 (3.6%) |
| Neighbours | 15 (2.2%) |
| Other | 4 (0.6%) |

**Table 6. Non-prescription antibiotics used by the respondents and symptoms treated with non-prescription antibiotics (n = 675).**

| Name of antibiotic | n = 675 | % |
|---|---|---|
| Azithromycin | 202 | 29.9% |
| Amoxicillin/clavulanic acid | 181 | 26.8% |
| Erythromycin | 87 | 12.9% |
| Ciprofloxacin | 43 | 6.4% |
| Cefixime | 31 | 4.6% |
| Metronidazole | 19 | 2.8% |
| Clarithromycin | 17 | 2.5% |
| Clindamycin | 15 | 2.2% |
| Cephalexin | 14 | 2.1% |
| Levofloxacin | 12 | 1.8% |
| Other | 24 | 3.6% |
| Unknown antibiotic | 195 | 28.9% |
| **Symptoms treated with non-prescription antibiotics.** | | |
| Respiratory tract symptoms. | 257 | 38.1% |
| Cough. | 205 | 30.4% |
| Common cold. | 177 | 26.2% |
| Abdominal pain. | 99 | 14.7% |
| Urinary tract symptoms. | 92 | 13.6% |
| Malaria. | 26 | 3.9% |
| Ear infections. | 24 | 3.6% |
| Sore throat. | 45 | 6.7% |
| Wounds | 8 | 1.2% |
| Other. | 30 | 4.4% |

## Discussion

Medical education before graduation about antimicrobial resistance is one of the important sources that increase their knowledge about antibiotics. Subjects such as pathology courses, pharmacology, clinical pharmacy, and microbiology are opportune for antibiotic and antimicrobial resistance education. The medical curricula of various universities that increases student knowledge and backgrounds about AMR was not available to be assessed. However, this study sought to investigate medical students' knowledge about antibiotics and antimicrobial resistance including the secondary aim to evaluate possible differences in their attitude and practice outcomes.

Our findings in this study showed that 329 (29.6%) and 484 (43.6%) of medical students appeared to have good and moderate knowledge about antibiotics respectively, while 297

**Table 7. Reasons for self-medicating with antibiotics (n = 675).**

| | n | % |
|---|---|---|
| Previous experience with similar symptoms | 335 | 49.6% |
| Costly doctor's appointments | 138 | 20.4% |
| No waiting | 177 | 26.2% |
| Non-expert recommendations | 31 | 4.6% |
| Leftover antibiotics at home | 61 | 9.0% |
| Other | 43 | 6.4% |

**Table 8. Practice of antibiotic use and storage of antibiotics (n = 1,110).**

| | Never n (%) | Sometimes n (%) | Always n (%) |
|---|---|---|---|
| **Practice** | | | |
| **I complete the course of antibiotics.** | 78 (7.0%) | 540 (48.6%) | 492 (44.3%) |
| **I carefully read the leaflet provided.** | 59 (5.3%) | 369 (33.2%) | 682 (61.4%) |
| **I experience effects when taking antibiotics.** | 587 (52.9%) | 442 (39.8%) | 81 (7.3%) |
| **When using intravenous antibiotics, I administer them myself.** | 978 (88.1%) | 108 (9.7%) | 24 (2.2%) |
| **Storage** | | | |
| **How often do you store antibiotics at home?** | 160 (14.4%) | 672 (60.5%) | 278 (25.1%) |

(26.7%) of medical students in this study had poor knowledge. Studies in southeast Asia and Italy have also shown significantly higher knowledge about antibiotics among final-year allied health students [26, 27]. It has also been stated that poor knowledge may lead to inappropriate antibiotics consumption which can result in a corresponding increase in bacterial resistance antibiotics [28].

Our study finds that there was a difference between medical students in clinical science years (seniors) and those in their initial basic science (juniors) years when it comes to their knowledge about antibiotic resistance. The results of this study identified that senior students have significantly better knowledge scores when compared to juniors (P < 0.001), which concurs with the findings in a study among pharmacy students in Sri Lankan universities [29]. Our results support that medical students from various areas, universities and backgrounds are aware about the growing global problem of antimicrobial resistance stated by the current evidence [28, 30]. A recent cross-sectional study in 29 European countries in 2018 discovered that medical students still need more education about the proper use of antibiotics for their future practice [31].

The majority (63.6%) held the misconception that antibiotics can be requested from healthcare providers during consultations. Patient request and coercion has proven to influence the prescription behaviour of healthcare providers, especially in the absence of continuing education and external responsibility. These factors greatly influence the dispensing behaviour among physicians, with malpractice being more common among healthcare providers expressing their refusal to communicate effectively with patients, in order to achieve time-effectiveness [10, 21, 22]. In Tanzania, 25.5% also preferred use of non-prescription antibiotics for minor illnesses as opposed to consulting doctors. This has been cited as a common reason for antibiotic self-medication in Namibia [32, 33].

Similar to a study conducted in UAE, a considerable proportion of final year medical students demonstrated a positive attitude to antibiotic use [26]. Academic seniority was also significantly associated with attitude scores (P < 0.01), with first-to-third year students achieving lower scores. Antibiotic resistance is not given sufficient cover in medical curricula in Sudan, while liberal practice of antibiotic use may be motivated by the acquisition of tuition in pharmacology and infectious diseases during this period. These findings are consistent with a previous study in China, where third-year medical students achieved the lowest attitude scores [34].

**Table 9. Self-medication and storage storage of antibiotics.**

| Are antibiotics stored in the home? | Total n (%) | Self-medicated with antibiotics n = 675 | P |
|---|---|---|---|
| **Yes** | 759 (68.4%) | 488 | < 0.001 |
| **No** | 351 (31.6%) | 187 | |

Our results in this study have shown that the prevalence of self-medication with antibiotics within the previous 12 months among medical students in Sudan is 60.8%, which is a very high rate. Such results are not uncommon in Sudan as similar trends have been observed in previous studies among undergraduate students and the general population in Khartoum State [10, 21, 22]. In Africa and the Eastern Mediterranean regions, this trend is also not unique to Sudan. Several studies on self-medication with antibiotics in both regions have portrayed similar and higher prevalence rates. In the African region, a study in northeastern Tanzania has shown that the prevalence of self-medication with antibiotics is 58%, while in Namibia, a study conducted on antibiotic self-mediation among children with acute respiratory infections depicted a prevalence rate of 60%. In rural Nigeria, a considerably higher prevalence rate (82.2%) was observed [32–35].

In the Eastern Mediterranean region, studies in UAE and Saudi Arabia reported prevalence rates of 52.1% and 78.8% respectively [36, 37].

Many significant predictors of self-medication among undergraduate medical students in Sudan were identified in this study. Females were significantly more likely to use non-prescription antibiotics than males in this study ($p < 0.01$). Previous studies in Khartoum State [10, 21], also reported self-medication with antibiotics being higher among females. This was also a common finding in multiple studies across Africa. In studies conducted among university students in Ethiopia and Nigeria [35, 38–40], female gender was also a significant determinant of self-medication with antibiotics. This could be due to the practice of sharing various types of medications being higher among females, although this requires confirmation through further studies.

Monthly allowance was also found to be a predictor of antibiotic self-medication in this study ($p = 0.011$). In lower-middle-income and low-income countries, low income may force individuals to search ways for decreased expenditure associated with access to healthcare, often by avoiding consultations with primary care and specialist physicians [41]. The development of health systems capable of providing healthcare to the population without financial hardship, in conjunction with a regulatory framework on antibiotic use would thus be crucial in combating the emergence of antibiotic-resistant strains resulting from the irrational use of antibiotics [42].

The results of this study indicated that the most commonly used antibiotics to practice self-medication were azithromycin (29.9%) and amoxicillin/clavulanic acid (26.8%). Amoxicillin/clavulanic acid was also among the most popular antibiotics used for self-medication in Ethiopia [38] and Eritrea [43]. Amoxicillin/clavulanic acid is widely prescribed by healthcare providers worldwide, inexpensive and is considered as first-line therapy in many lower-middle-income and low-income countries where infectious diseases are highly prevalent [44]. In this study, azithromycin was the most popular antibiotic—a finding which was not observed in previously conducted studies in Sudan [10, 21, 22, 44] This change may be due to the resistance of bacterial strains as a result of over-prescription and misuse amoxicillin/clavulanic acid. Amoxicillin/clavulanic acid is commonly sold in pharmacies as Amoclan in Sudan. Amoclan is effective in treatment of infections caused by Klebsiella and Escherichia coli. However, resistant strains have emerged as a result of its overuse [45, 46]. This may also be explained by clients and patients requesting more potent antibiotics from their pharmacies. The use of higher-potency antibiotics also suggests that pharmacists and physicians may inform their patients about infections resistant to certain antibiotics, while patients use this information independently after their consultations. Azithromycin was also commonly prescribed in Mozambique and India [47, 48]. 195 respondents used unknown antibiotics, with unknown dosage and course durations. Incomplete courses and indiscriminate drug use increase the

risk of side effects rom polypharmacy and the emergence of antibiotic-resistant strains [10, 49, 50].

The most common symptoms treated with non-prescription antibiotics were acute respiratory tract symptoms (38.1%), cough (30.4%) and common cold (26.2%). These ailments do not usually require antibiotic treatment and the practice of non-prescription antibiotic use for their treatment is a global commonality [41, 51–59]. In accordance with the response of the World Health Organization to the global crisis of antimicrobial crisis, a universal protocol regarding antibiotic prescription is adopted by clinicians. Bacteria are the only class of organisms against which antibiotics have any effect, with infection by different strains of bacteria requiring treatment with different antibiotics. It is on these grounds that only a fully qualified healthcare provider is given the right to decide which antibiotic is used in particular bacterial infections, after thorough clinical assessment of each patient, in accordance which local and national guidelines. These policies promote patient safety and safer, more economical utilisation of antibiotics among both prescribers and patients [60–62]. If the public were to be aware of these policies, the practice of self-medication with antibiotics would likely be discouraged. A one-health approach should therefore be applied in an effort to raise awareness and share information regarding the correct protocol of antibiotic use, which would lead to the making of more informed decisions regarding the use of non-prescription antibiotics.

Of the 675 respondents who practiced non-prescription antibiotic use, 47.5% sought these medications from community pharmacies, while 9.0% used leftover antibiotics. This pattern remains consistent in Sudan [10, 21, 22, 44] and thus demonstrates laxity within its health system in prohibition of sales of antibiotics without valid prescriptions. Similar experiences also motivated 49.6% to use nonprescription antibiotics as a cost-effective option. This trend can be combated by enacting policies prohibiting the inappropriate sales in private pharmacies and maintaining supplies in hospital pharmacies and public health facilities. In low-income countries, low supplies of essential drugs in hospital pharmacies may drive patients to pay higher prices in private pharmacies [63]. The enactment of policies prohibiting irrational sales of antibiotics can lead to considerably lower antibiotic self-mediation rates, thus proving beneficial in combating antimicrobial resistance [64, 65]. Leftover antibiotics are among the most frequently stored antibiotics in low-income countries, indicating frequent non-compliance with the course of anti-infective treatment in conjunction with an increased risk of sharing [66].

The authors acknowledge that this study is not without limitations. As this is a retrospective questionnaire-based study, the results of the study are entirely dependent on the responses given by the participants and are therefore influenced by recall bias. Also, the questionnaire itself was administered online due to high travel expenses amid a difficult political climate. Despite these barriers, given the applied sampling technique, the authors believe that the results give an appropriate portrayal of knowledge and attitude towards antibiotic use and antibiotic resistance, paired with an accurate estimate of self-medication with antibiotics among medical students in Sudan.

## Conclusions

In this study, undergraduate medical students in Sudan demonstrated moderate levels of knowledge and attitude towards antibiotics. The prevalence of self-medication with antibiotics is very high among Sudanese medical students, with community pharmacies being a popular source of non-prescription antibiotics. Policy reform and legislation to regulate sales of antibiotics as prescription-only medications and revision of content on antimicrobial resistance in medical curricula in Sudan are therefore crucial in addressing antibiotic misuse and improving knowledge and attitude towards antibiotic use among medical students.

## Acknowledgments

We acknowledge all the respondents for their time in participating in this study.

## Author Contributions

**Conceptualization:** Osman Kamal Osman Elmahi, Reem Abdalla Elsiddig Musa.

**Data curation:** Osman Kamal Osman Elmahi, Reem Abdalla Elsiddig Musa, Ahd Alaaeldin Hussain Shareef, Mohammed Eltahier Abdalla Omer, Tagwa Faisal Mohamed Alsadig.

**Formal analysis:** Osman Kamal Osman Elmahi, Reem Abdalla Elsiddig Musa, Mugahid Awad Mohamed Elmahi, Randa Ahmed Abdalrheem Altamih, Tagwa Faisal Mohamed Alsadig.

**Investigation:** Ahd Alaaeldin Hussain Shareef, Mugahid Awad Mohamed Elmahi, Rayan Ibrahim Hamid Mohamed.

**Project administration:** Osman Kamal Osman Elmahi, Reem Abdalla Elsiddig Musa.

**Supervision:** Tagwa Faisal Mohamed Alsadig.

**Validation:** Osman Kamal Osman Elmahi, Reem Abdalla Elsiddig Musa, Ahd Alaaeldin Hussain Shareef, Mohammed Eltahier Abdalla Omer, Mugahid Awad Mohamed Elmahi, Rayan Ibrahim Hamid Mohamed, Tagwa Faisal Mohamed Alsadig.

**Visualization:** Ahd Alaaeldin Hussain Shareef, Tagwa Faisal Mohamed Alsadig.

**Writing – original draft:** Osman Kamal Osman Elmahi, Reem Abdalla Elsiddig Musa, Ahd Alaaeldin Hussain Shareef, Mohammed Eltahier Abdalla Omer, Mugahid Awad Mohamed Elmahi, Randa Ahmed Abdalrheem Altamih, Rayan Ibrahim Hamid Mohamed, Tagwa Faisal Mohamed Alsadig.

**Writing – review & editing:** Osman Kamal Osman Elmahi, Reem Abdalla Elsiddig Musa, Ahd Alaaeldin Hussain Shareef, Mohammed Eltahier Abdalla Omer, Mugahid Awad Mohamed Elmahi, Randa Ahmed Abdalrheem Altamih, Rayan Ibrahim Hamid Mohamed, Tagwa Faisal Mohamed Alsadig.

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
