## [Decision Letter · Decision Letter 0]

7 Nov 2021

PONE-D-21-20765Perception and practice of self-medication with antibiotics among medical students in Sudanese universities: a cross-sectional study.PLOS ONE

Dear Dr. Elmahi,

Thank you for submitting your manuscript to PLOS ONE. After careful consideration, we feel that it has merit but does not fully meet PLOS ONE’s publication criteria as it currently stands. Therefore, we invite you to submit a revised version of the manuscript that addresses the points raised during the review process. Please submit your revised manuscript by Dec 22 2021 11:59PM. If you will need more time than this to complete your revisions, please reply to this message or contact the journal office at plosone@plos.org. Please include the following items when submitting your revised manuscript:A rebuttal letter that responds to each point raised by the academic editor and reviewer(s). You should upload this letter as a separate file labeled 'Response to Reviewers'.A marked-up copy of your manuscript that highlights changes made to the original version. You should upload this as a separate file labeled 'Revised Manuscript with Track Changes'.An unmarked version of your revised paper without tracked changes. You should upload this as a separate file labeled 'Manuscript'.If applicable, we recommend that you deposit your laboratory protocols in protocols.io to enhance the reproducibility of your results. Protocols.io assigns your protocol its own identifier (DOI) so that it can be cited independently in the future. For instructions see: https://journals.plos.org/plosone/s/submission-guidelines#loc-laboratory-protocols. Additionally, PLOS ONE offers an option for publishing peer-reviewed Lab Protocol articles, which describe protocols hosted on protocols.io. Read more information on sharing protocols at https://plos.org/protocols?utm_medium=editorial-email&utm_source=authorletters&utm_campaign=protocols.

We look forward to receiving your revised manuscript.

Kind regards,

Xiaoxv Yin

Academic Editor

PLOS ONE

2. Please change "female” or "male" to "woman” or "man" as appropriate, when used as a noun (see for instance https://apastyle.apa.org/style-grammar-guidelines/bias-free-language/gender).

Reviewers' comments:

Reviewer's Responses to Questions

**Comments to the Author**

1. Is the manuscript technically sound, and do the data support the conclusions?

Reviewer #1: Partly

Reviewer #2: Partly

2. Has the statistical analysis been performed appropriately and rigorously? 

Reviewer #1: Yes

Reviewer #2: Yes

3. Have the authors made all data underlying the findings in their manuscript fully available?

Reviewer #1: No

Reviewer #2: Yes

4. Is the manuscript presented in an intelligible fashion and written in standard English?

Reviewer #1: Yes

Reviewer #2: Yes

5. Review Comments to the Author

Reviewer #1: Please add the validity and reliability of the questionnaire.

Please synchronize information regarding data analysis in the abstract, methods, and results.

Please mention the response rate of the participants.

Reviewer #2: I would like to express my gratitude for the invitation to review this manuscript. This study was conducted to reveal the factors that affect the self-medication with antibiotics among medical students in Sudanese universities. I have some suggestions to improve the manuscript.

1. Please add the current status and limitations of relevant research in the introduction section.

2. It is good to comment on how the questionnaire was developed, e.g. based on previous publications, etc., and was any pilot undertaken to enhance its robustness?

3. Data Analysis part needs further improvement. Why is there no mention of logistics Logistic regression?

6. PLOS authors have the option to publish the peer review history of their article (what does this mean?). If published, this will include your full peer review and any attached files.

Reviewer #1: No

Reviewer #2: No

---

## [Author Response · Author response to Decision Letter 0]

31 Dec 2021

Reviewer #1:

The data analysis information written in the abstract is written in tandem with the order present in the data analysis part in the methodology section of the manuscript. 

Reviewer #2:

The questionnaire was pretested and further developed for robustness following a pilot test. The results of the pilot test were used to arrive at the prevalence rate of antibiotic self-medication implemented in the sampling equation used to calculate the sample size in this study.

---

## [Editor Report · Decision Letter 1]

12 Jan 2022

Perception and practice of self-medication with antibiotics among medical students in Sudanese universities: a cross-sectional study.

PONE-D-21-20765R1

Dear Dr. Osman Kamal Osman Elmahi,

We’re pleased to inform you that your manuscript has been judged scientifically suitable for publication and will be formally accepted for publication once it meets all outstanding technical requirements.

Kind regards,

Xiaoxv Yin

Academic Editor

PLOS ONE
---

## [Editor Report · Acceptance letter]

17 Jan 2022

PONE-D-21-20765R1 

Perception and practice of self-medication with antibiotics among medical students in Sudanese universities: a cross-sectional study. 

Dear Dr. Elmahi:

I'm pleased to inform you that your manuscript has been deemed suitable for publication in PLOS ONE. Congratulations! Your manuscript is now with our production department. 

Kind regards, 

on behalf of

Dr. Xiaoxv Yin 

Academic Editor

PLOS ONE